# Association between Microbiota and Nasal Mucosal Diseases in terms of Immunity

**DOI:** 10.3390/ijms22094744

**Published:** 2021-04-29

**Authors:** Junhu Tai, Mun Soo Han, Jiwon Kwak, Tae Hoon Kim

**Affiliations:** Department of Otorhinolaryngology-Head & Neck Surgery, College of Medicine, Korea University, Seoul 02841, Korea; junhu69@korea.ac.kr (J.T.); mshan35@gmail.com (M.S.H.); jwon1111@naver.com (J.K.)

**Keywords:** microbiota, nasal mucosa, immunity, chronic rhinosinusitis, allergic rhinitis

## Abstract

The pathogenesis of nasal inflammatory diseases is related to various factors such as anatomical structure, heredity, and environment. The nasal microbiota play a key role in coordinating immune system functions. Dysfunction of the microbiota has a significant impact on the occurrence and development of nasal inflammation. This review will introduce the positive and negative roles of microbiota involved in immunity surrounding nasal mucosal diseases such as chronic sinusitis and allergic rhinitis. In addition, we will also introduce recent developments in DNA sequencing, metabolomics, and proteomics combined with computation-based bioinformatics.

## 1. Introduction

In 2020, a panel of international experts, with more than 100 representatives from all over the world, defined the term “microbiota” [1]. They concluded that microbiota are composed of prokaryotes and eukaryotes, and that they are active within the range of microbial structures, metabolites, and movable genetic elements. Whipps et al. [2] first defined the term “microbiome” in 1988. Microbiomes encompass a wider range than microbiota. Phages, viruses, plasmids, free DNA, and a number of others such as prions and viroids are not considered to be members of the microbiota, but are included in the microbiome [3]. 

Many studies were conducted regarding gut microbiota, involving the most intensive and diverse microbial communities, which represent approximately 1.5 kg of bacteria in the gut [4,5]. pH of the gut, bile acids, and components of the innate immune system operate together to select and identify members of the gut microbiota [6,7,8]. Gut microbiota play a significant role in four domains of the human body. First, with regard to metabolism, gut microbiota can use dietary fiber and undigested proteins to provide energy and a variety of nutrients for the human body. Second, gut microbiota can protect the gut by secreting antimicrobial peptides, secretory IgA, and short-chain fatty acids. Third, they can also upregulate the expression of tight junction proteins, thereby improving the gut structure. Fourth, the gut is connected to the brain through the enteric nervous system (ENS), and neurotransmitters produced by gut microbiota participate in a variety of nervous systems [9]. Recently, various studies reported that changes in the gut microbiota are related to multiple disorders such as inflammatory bowel disease [10], Crohn’s disease [11], hepatitis C [12], Alzheimer’s disease [13], and depression [14].

Although the relevant mechanisms are still being studied, increasing evidence emphasizes the effect of gut microbiota on lung immunity, which is referred to as the gut-lung axis [15]. Bacterial community in the lungs is similar with that in the mouth. *Streptococcus*, *Prevotella*, and *Veronica* are the most commonly encountered genera [16]. Components of the pulmonary microbiota are thought to be transported from the oropharynx through micro-inhalation events and by mucosal diffusion involving adjacent tissues [17]. Various studies showed that changes in microbiota have a certain impact on the immunity of the lower airway mucosa. Eosinophilic inflammation, *TH17* gene expression, neutrophilic inflammation, and the markers of allergic inflammation are all related to the differences in airway microbiota composition [18]. A report concluded that *Staphylococcus*, *Propionibacterium*, *Corynebacterium*, and *Streptococcus* are common bacterial genera in the nasal mucosal diseases of chronic rhinosinusitis (CRS) and allergic rhinitis (AR) [19]. In the upper respiratory tract immune system, microbiota are tolerated due to the low reactivity of the host immune system [20], and dysbiosis in the microbiome results in diseases of the upper respiratory tract, similarly to that in other human body sites [21].

Human nasal mucosa is the first contact point of inhaled environmental insults. Just as gut microbiota can protect the intestinal mucosa through immune regulation, microbiota in nasal mucosa are likely to play an important role in mucosal immunity. Although much research was conducted concerning the role of microbiota in lower respiratory tract disorders such as asthma, the role of microbiota in the upper respiratory tract including human nasal mucosa has not been studied in detail, especially with respect to immunity. Therefore, this review aims to clarify the relationships between different types of nasal mucosal diseases and microbiota in immunity, and introduces new technologies and methods to study microbiota.

## 2. CRS and Microbiota

### 2.1. CRS Classification

CRS is a chronic inflammatory disease that occurs in the nasal cavity and sinuses and affects 12% of the global population [22]. The phenotypic classification of CRS is mainly based on the presence or absence of nasal polyps, which can be divided into CRS with nasal polyps (CRSwNP) or CRS without nasal polyps (CRSsNP) [23]. In contrast to the phenotype, the endotypic classification of CRS mainly represents an individual’s inflammatory mechanisms, rather than a clear entity with a direct biological basis. In a very meaningful study, researchers used 14 different inflammatory markers for hierarchical cluster analysis to determine the putative inflammatory endotype of CRS, and identified ten different clusters, including eosinophils and T helper-(Th) 2 related markers such as interleukin (IL)-5 and immunoglobulin E (IgE), neutrophils, or proinflammatory mediators such as IL-1β, IL-6, IL-8, and myeloperoxidase; Th17/Th22 markers such as IL-17A, IL-22, and tumor necrosis factor-α (TNF-α); and interferon-γ (IFN-γ) [24]. In recent years, a deeper understanding of the role of microbiota in the human immune system evolved; various inflammatory diseases, such as CRS, were reported to be associated with a significant shift in host microbiota from a healthy state to a diseased state [25]. Compared with that on other disorders such as asthma, research concerning microbiota and nasal diseases is still in its infancy, and causal relationships involving the existence of microbial communities and the development of CRS cannot be readily explained [26]. 

There are different types of microbiota in the upper airway of healthy adults (Figure 1). The nasal cavity is directly connected to the external environment. Through inhalation, the nasal cavity can directly contact various microbiota, fungal spores, and pollutants [27]. The microbiota of healthy adults’ anterior nares is mainly composed of *Actinobacteria*, *Firmicutes*, and *Proteobactera* [28]. Researchers examined the anterior nares of 236 healthy adults using nasal swabs and concluded that *Staphylococcus*, *Propionibacterium*, *Corynebacterium*, and *Moraxella* were the most common microbiota in their anterior nares [29]. One study concluded that tissue samples were more suitable for assessing microbiological groups in CRS patients than nasal swabs, because they observed significant differences in the microbiota groups in the nasal swabs, while the differences observed in the tissue samples were smaller [30]. However, there are some discrepancies about the usefulness of the two methods. [31]. Other data showed that a tissue biopsy cannot provide additional information compared with multiple swab tests. In more than 90% of their cases, swabs from multiple sites provide comprehensive information about patients’ culturable pathogens. In the middle meatus of healthy adults, the most abundant microbiota were *Staphylococcus aureus* (*S. aureus*), *Staphylococcus epidermidis*, and *Propionibacterium acnes* [32]. Using next-generation 454 pyrosequencing of the 16S rRNA gene, Jetté et al. found that *Streptococcus*, *Prevotella*, *Veillonella*, and *Haemophilus* were the most common microbiota in the throats of 97 adults [33]. The common microbiota in CRS patients vary with geographical location. According to data reported, *Cyanobacteria* are the dominant phylum in CRS in Missouri, USA, and that the change in microbiota composition between the control group and CRS group is minor [34]; in contrast, other data show that the abundance of *Verrucomicrobia* and *Bacteroides* is low and that of *Actinobacteria* is high in Colorado, USA [35]. A Korean study compared a CRS group with a control group and found that the abundance of *Bacteroides* in the CRS group was low and that of *Fusobacteria* was high [36].

### 2.2. Type 2 CRS and Microbiota

Relationships between CRS and various microbiota cultured in the nasal system have been studied for many years [37]. Microbiotal dysbiosis is considered an important biomarker of CRS [38,39]. Recent advances in new detection methods aroused interest in the role of microbiota in long-term diseases, which can be used to identify previously unrecognizable, unculturable microbiota. Quantitative polymerase chain reaction (qPCR), fluorescence in situ hybridization (FISH), mass spectrometry, and DNA microarrays were used to identify microbiota and to visualize biofilms in clinical samples of patients with CRS [40]. The development of next-generation sequencing (NGS) provides a non-targeted molecular method. Specifically, 16S amplicon DNA sequencing is a NGS technique in which universal primers for the 16S rRNA gene are used; this suggests that bacterial organisms participate in the pathogenesis of CRS, and may indicate that disordering of normal microbiota community structures in the nasal sinus mucosa is one of the causes of CRS [41,42]. Metabolic exchange plays an important role in maintaining the interdependence between microbiota [43]. An outstanding study pointed out that *Corynebacterium*, one of the common nasal bacteria, inhibits the growth of *Streptococcus pneumoniae* by releasing triacylglycerol on the skin surface of the host [44]. Some studies also found that in CRS, the growth of *S. aureus* is often closely related to *Staphylococcus epidermidis* and *Propionibacterium acnes* [45]. A most recent study in South Korea obtained interesting data [46]. They found that the use of antibiotics can cause differences in secretory proteome according to the condition of the disease. Their data suggest that the use of antibiotics should be considered as a confounding factor in proteomics research.

There are various microbiota, such as *Staphylococcus*, *Streptococcus*, *Propionibacterium*, and *Corynebacterium*, (Figure 2A) in normal nasal mucosa [47]. However, the type and quantity of microbiota change significantly in the mucosa of patients with CRS (Figure 2B,C). Changes in microbiota are related to various factors. In addition to significant differences between subjects, age and smoking influence the composition and distribution of microbiotal species [48,49]. The frequent use of antibiotics may also induce instability in the microbiota. A previous study compared the microbiota of paranasal sinuses of patients with CRS before and after drug treatment and found that the diversity and uniformity of bacteria decreased significantly after high-dose antibiotic treatments [50]. In a cross-sectional study, surgery was shown to affect the microbiotal ecology of the sinuses, leading to a reduction in microbiota abundance [35]. Sinus surgery has a similar effect on fungal populations. The abundance and diversity of fungi in the sinus cavity of patients after endoscopic sinus surgery are significantly decreased [51,52]. The results of a meta-analysis [53] revealed that bacterial richness and diversity in CRS decreased, which supports the keystone-pathogen hypothesis; that is, certain pathogenic microbiota that usually exist in low abundance may form a microbiome under disease conditions [54]. Disruption of the microbial community leads to the loss of key symbiotic species. Under normal circumstances, these symbiotic species may prevent the excessive growth of pathogens, and the loss of variety and diversity of CRS microbiota seems to be the products of tissue eosinophilia and mucosal inflammation; whether this disorder is a cause or a consequence of an impaired epithelial integrity disease remains a subject for further research [55].

*S. aureus* is a common type of microbiota in the nasal mucosa [56], but many studies have shown that compared with that in the normal nasal mucosa, the number of *S. aureus* in the patients with type 2 CRS is greatly increased (Figure 2B) [57]. *S. aureus* produces enterotoxins, and it can be recognized as superantigens and by Th-2 inflammation that is promoted by it; *S. aureus* also leads to the secretion of cytokines such as IL-13, IL-4, and IL-5 in type 2 CRS [58]. Mucosal ulceration is associated with an increased abundance of *Bacteroides* [59], squamous metaplasia associated with high *Streptococcus* levels [60]; *Prevotella* is related to the release of proinflammatory cytokines [61], and the production of thymic stromal lymphopoietin(TSLP) is related to the induction of fungal protease, which leads to the activation of type 2 innate lymphoid cells (ILC2s) producing IL-5 and IL-13 [62].

### 2.3. Non-Type 2 CRS and Microbiota

Non-type 2 CRS is a heterogeneous disease; additional definitions are needed to guide its correct diagnosis and treatment, as most CRS studies focus on type 2 CRS, and there is not enough information regarding the internal classification of non-type 2 CRS [63]. Diseases that can easily induce non-type 2 CRS include acute rhinosinusitis, which is usually caused by a viral respiratory tract infection, asthma, tonsillitis, bronchitis, allergic and non-allergic rhinitis, pneumonia, and gastroesophageal reflux disease [64]; however, potential susceptibility conditions, such as primary and secondary immunodeficiency, including HIV infection, cystic fibrosis, and cilia dyskinesia, should also be considered [65]. 

The number of eosinophils and plasma cells in the mucosa of non-type 2 CRS is less than that in type 2 CRS, but the number of neutrophils in the mucosa of non-type 2 CRS is higher than that in type 2 CRS (Figure 2C). Neutrophil inflammation of nasal mucosa is a characteristic of non-type 2 CRS, which is caused by infection or external stimuli; type 1 inflammation, which is based on the Th1 cell, and type 3 inflammation, which is based on the Th17 cell, are present in an equal proportion in non-type 2 CRS [66]. Invasion by external pathogens induces the secretion of IL-6, IL-8, and TNF-α in the nasal epithelium (Figure 2C), which could be caused by a rhinovirus [67] that activates dendritic cells (DCs). IL-8 secreted by epithelial cells recruits neutrophils, which cause goblet cells to proliferate and destroy tight junctions [68]. In 28 patients with asthma, terminal restriction fragment length polymorphism (T-RFLP) analysis showed that *Moraxella*, *Haemophilus*, and *Streptococcus* were the dominant species in the respiratory tract bacterial community, and the total abundance of these microbiota was significantly and positively correlated with the concentration of IL-8 and neutrophil count in sputum [69]. Lal et al. [47] conducted an inter-subject microbiotal analysis of 65 subjects. They found that the diversity of microbiota in patients with non-type 2 CRS was lower than that in the control group (healthy and AR subjects) or in patients with type 2 CRS. *Fusobacterium*, *Propionibacterium*, *Haemophilus*, and *Streptococcus* were the main bacteria in non-type 2 CRS patients. Therefore, it can be speculated that *Haemophilus* and *Streptococcus* may be involved in the secretion of IL-8 and recruitment of neutrophils in non-type 2 CRS (Figure 2). The reported data show that levels of IL-22 receptors are increased in non-type 2 CRS [70], and studies showed that IL-22 production in the gut is induced by *Clostridium* [71]. A combination of data from 51 patients with CRS suggested that the increase in IL-22 receptor levels in non-type 2 CRS may be a result of the predominance of *Clostridium* in nasal microbiota and IL-22 cytokine production [60]. 

## 3. AR and Microbiota

### 3.1. AR and Type 1 Hypersensitivity

AR is a common symptom of type 1 hypersensitivity and Th2-mediated inflammatory disease [72]. Epidemiological studies show that nearly a quarter of adults and almost half of children are affected [73]. AR was previously considered to be a disease confined to the nasal cavity; however, it is now considered to be a manifestation of systemic airway disease, which is usually comorbid with asthma [74]. As an IgE mediated type 1 hypersensitivity process, AR symptoms are caused by allergens, and when the nasal mucosa is directly or indirectly exposed to allergens such as mold, pollen, dust, and mite feces, innate immune cells and adaptive immune cells participate in the pathophysiological mechanisms involved in AR, inducing IgE production, eosinophil activation, mast cell recruitment, and basophils degranulation, and then present a variety of clinical symptoms of AR [75]. Therefore, minimizing allergen exposure should be an important part of any treatment plan [76]. In addition to avoiding known allergens, intranasal corticosteroids, which are one of the most effective therapeutics, should be used as the first-line treatment; however, when there is no response to intranasal corticosteroids, second-line treatment should be considered, including antihistamines, decongestants, leukotriene receptor antagonists, and non-drug treatments, such as nasal irrigation. Subcutaneous or sublingual immunotherapy should be considered if a patient’s AR symptoms cannot be fully controlled by conventional treatment modalities [77].

As time goes by, more and more achievements were made in elucidating the occurrence and mechanisms underlying anaphylaxis. The key processes involved in anaphylaxis include the activation and maturation of DCs after exposure to allergens; subsequently, the initial signals provided by mucosal epithelial cells and DCs lead to the cloning and expansion of allergen-specific Th2 cells, which are important driving factors in AR pathology [78]; it was shown that Th2 cells are related to the sensitization and staging of AR [79]. ILC2s are also activated by cytokines such as IL-25, IL-33, and TSLP. Th2 cells and ILC2s produce type 2 cells, including IL-4, IL-5, IL-13, IL-25, IL-33, and TSLP. IL-4 and IL-13 drive B cells to produce allergen-specific IgE, which can combine with mast cells. IL-5 contributes to eosinophil recruitment of eosinophils [80]. Some studies reported that serum IL-17 level is correlated with allergic severity during high pollen-level seasons, which is considered to be a marker of the severity of the allergy in patients with AR. Additionally, myeloid DCs isolated from patients with pollen allergy increase the tendency of inducing T cells to secrete IL-17 in vitro [81]. Viral infection may lead to the occurrence and aggravation of AR. During the common cold, mast cells congregate, leading to deterioration in allergic conditions; the key factors influencing such allergic reactions are stage, genetic background, gender, and age at viral infection [82]. 

### 3.2. AR and Microbiota

The incidence rate of allergic diseases is closely related to interactions between the host system and resident microbiota [83]. Some studies showed that symbiotic microbiota regulate susceptibility to allergic diseases, and the absence of symbiotic bacteria can enhance the proliferation of basophils, increase the number of infiltrating lymphocytes and eosinophils, aggravate Th2 cell reactions and allergic inflammation, and reduce the number of regulatory T (Treg) and Th17 cells [84,85].

This situation is similar to that encountered in normal sinus mucosa; *S. aureus*, *Propionibacterium*, *Prevotella*, *Corynebacterium*, *Bacteroidetes*, and *Streptococcus* are common in normal nasal mucosa (Figure 3A); however, the abundance of *S. aureus*, *Propionibacterium*, *Corynebacterium*, and *peptoniphilus* in the nasal mucosa of patients with AR is considerably increased compared with that of the common bacteria in nasal mucosa of normal individuals (Figure 3B), while the number of *Prevotella* and *Streptococcus* is decreased [47]. In a study of 20 patients with AR and 12 normal controls, the researchers used 454 pyrosequencing based on the 16S rRNA gene to describe and compare the inferior turbinate mucosal microbiota of normal controls and patients with AR, and found that the inferior turbinate microbiota imbalance in patients with AR was related to the total IgE level; their results emphasized the relationship between inferior turbinate microbiota imbalance and the onset of AR [86].

In the upper respiratory tract, microbiota play an important role in driving type 2 immune responses, according to the data of a study published in 2016; after binding with toll like receptor (TLR) 2, *S. aureus* induces the production of type 2 cytokines, such as IL-5 and IL-13, via IL-33 released from human airway epithelial cells and TSLP [87]. Furthermore, staphylococcal enterotoxin B (SEB) induces IL-5 and IL13 release by affecting Th2 cells [88]. Rhinovirus, one of the most common viruses in the human respiratory tract, is closely related to the occurrence and development of allergic asthma and plays a key role in the propagation of the type 2 immune response. IL-25 and IL-33 are produced by human respiratory epithelial cells stimulated by rhinovirus, which then drive the production of IL-5 and IL-13 by binding to the receptors on Th2 cells, ILC2s, and basophils [89,90]. These released type 2 cytokines are actively involved in the type 2 immune response. IL-5 participates in the recruitment of eosinophils and is related to their development and activation [91]. IL-13 upregulates class II expression in B cells and promotes IgE class conversion, and then IgE binds to mast cell receptors [92].

Although there are relatively few studies regarding co-infection involving viruses and bacteria, they may involve some special reactions. Some studies reported that co-infection with viruses and bacteria increases the risk of rehospitalization in patients with asthma [93]. Exposure of human epithelial cells to *Haemophilus influenzae* significantly enhances the combination of epithelial cells and rhinovirus [94]. Children are more likely to experience severe airway inflammation when infected with a combination of *Mycoplasma pneumoniae* and a virus than when infected with a virus alone [95]. These observations suggest that a host’s defense ability may be decreased to varying degrees after combined infection with a variety of viruses or bacteria. It is imperative to understand the composition of microbiota in allergic diseases and the changes in immune status after infection further.

## 4. Conclusions

Microbiota play a complex role in immunity against CRS, non-type 2 CRS, and AR. Superantigens secreted by *S. aureus* and fungal proteases can lead to the release of a variety of interleukins in type 2 CRS, and *Prevotella* can induce the release of proinflammatory factors. Although the mechanism is not clear, the involvement of rhinovirus, *Haemophilus*, and *Streptococcus* in the immune process of non-type 2 CRS can be predicted. Similarly to that in type 2 CRS, *S. aureus* can induce the differentiation of dendritic cells and release interleukin in AR, while rhinovirus may participate in immune responses by basophils and Th2 cells. 

In conclusion, it can be assumed that changes in microbiota play a role in the induction of upper airway diseases; however, further research is needed to clarify the roles of various microbiota in the immune processes involved.

## Figures and Tables

**Figure 1 ijms-22-04744-f001:**
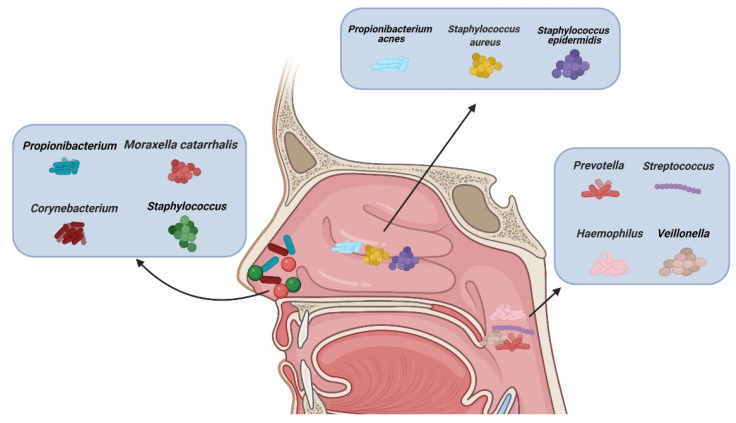
Types of microbiota in normal nasal mucosa and other parts of the upper airway. (Figure created with Biorender.com).

**Figure 2 ijms-22-04744-f002:**
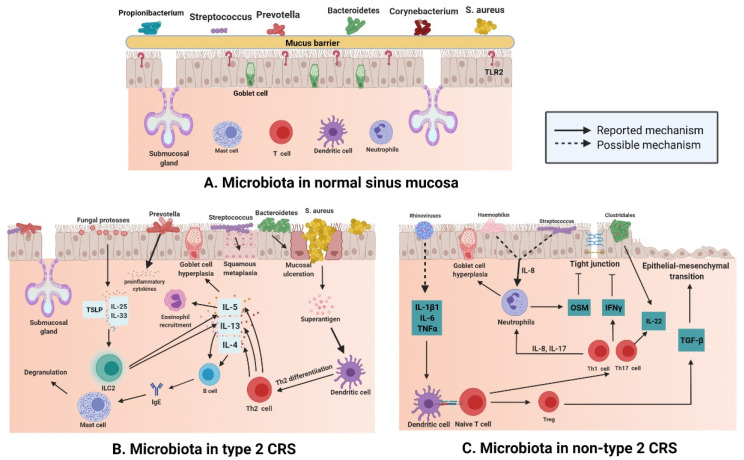
(**A**) Different kinds of microbiota colonize healthy nasal mucosa such as *Staphylococcus*, *Streptococcus*, *Propionibacterium*, and *Corynebacterium*. (**B**) With the loss of epithelial integrity, pattern recognition molecules decrease, which provides an environment for bacteria to enter through the mucosal barrier in type 2 CRS. Mucosal ulceration is associated with increased abundance of *Bacteroides*, while squamous metaplasia is associated with increased *Streptococcus* abundance; enterotoxins produced by *S. aureus* can act as superantigens and promote Th-2 inflammation, thereby leading to the production of cytokines, such as IL-13, IL-4, and IL-5, in Type 2 CRS. At the same time, *Prevotella* is related to the release of proinflammatory cytokines, and fungal proteases can induce the production of TSLP, which leads to the activation of ILC2s producing IL-5 and IL-13. (**C**) In non-type 2 CRS, an increase in the abundance of *Haemophilus* or *Streptococcus* may be related to elevated IL-8 levels and neutrophil counts. Rhinovirus may increase the levels of IL-1B1, IL-6, and TNF-α; induce dendritic cell differentiation; and boost IL-22. At the same time, increased abundance of *Clostridiales* also elevates IL-22 levels. (Figure created with Biorender.com).

**Figure 3 ijms-22-04744-f003:**
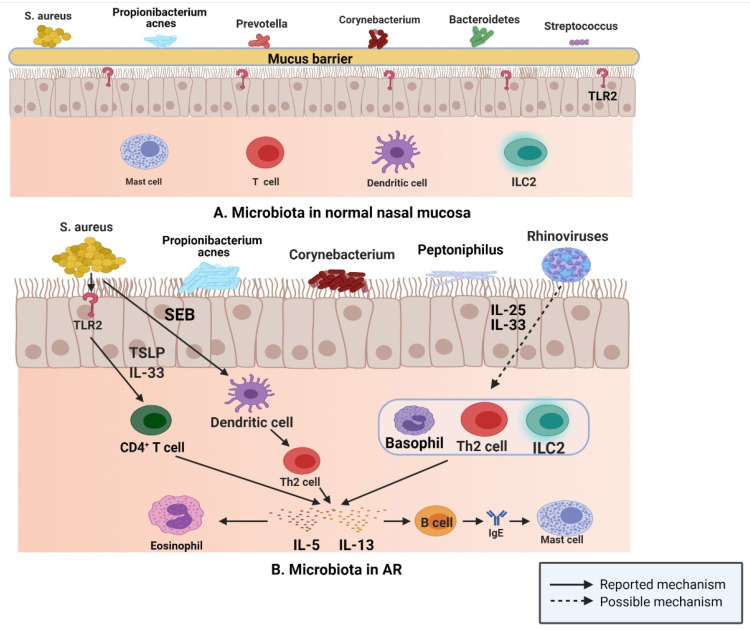
(**A**) *S. aureus*, *Propionibacterium*, *Prevotella*, *Corynebacterium*, *Bacteroidetes*, and *Streptococcus* are common in normal nasal mucosa. (**B**) In patients with AR, *S. aureus* can produce IL-5 and IL-13 by binding to TLR2, and SEB can induce the release of IL-5 and IL-13 by affecting Th2 cells. Rhinovirus stimulates human respiratory epithelial cells to produce IL-25 and IL-33, which drive the production of IL-5 and IL-13 by binding to Th2 cells, ILC2s, and basophils (Figure created with Biorender.com).

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
