# Peer review of "Association between Microbiota and Nasal Mucosal Diseases in terms of Immunity"

_ijms, 2021, doi:10.3390/ijms22094744_

Round 1

Reviewer 1 Report

Dear Authors,

The topic of the review is interesting and it handle well the immunity related to nasal mucosal diseases. Nevertheless, you write that you’ll introduce recent developments in DNA sequencing, metabolomics and so on but there are poor references about these topics. Please enforce the informations about metabolomics and proteomics.
Some suggestions:

Line 22: “a number of others” please specify what are others
Line 35: “a variety of nervous systems” I suppose you have miss a word
Line 48: I suggest to replace “microbiomes” with bacterial genera
Line 91: I suggest to insert “with geographical location”
From line 183 to line 187: this sentence isn’t clear, please rewrite
Figure 1: Propionibacterium acnes and Haemophilus haven’t a artwork, why? If it is possible add a picture or similar
Lines 84, 85, 87, 88, 89, 91, 94, 96, 115, 116, 137, 138, 139, 141, 142, 143, 180, 181, 186, 187, 188, 191, 193, 227, 238, 239, 240, 241, 243, 252, 267, 269, 270: pay attention to put all the bacteria in italic

Author Response

Point 1: Please enforce the information about metabolomics and proteomics

Response 1:

Thanks for your comments and we added the information about metabolomics and proteomics as follows:

Line 125

Metabolic exchange plays an important role in maintaining the interdependence between microbiota [43]. An outstanding study has pointed out that Corynebacterium, one of the common nasal bacteria, inhibits the growth of Streptococcus pneumoniae by releasing triacylglycerol on the skin surface of the host [44]. Some studies have also found that in CRS, the growth of S. aureus is often closely related to Staphylococcus epidermidis and Propionibacterium acnes [45]. A most recent study in South Korea has obtained interesting data [46]. They found that the use of antibiotics can cause differences in secretory proteome according to the condition of the disease. Their data suggest that the use of antibiotics should be considered as a confounding factor in proteomics research.

Point 2: Line 22: “a number of others” please specify what are others

Response 2:

Thank you for your comments, we have revised the manuscript as follows:

Line 22

and a number of others like prions and viroids

Point 3: Line 35: “a variety of nervous systems” I suppose you have missed a word

Response 3:

Thanks for the comments, we revised the manuscript by adding a word “which”.

Line 35

produced by gut microbiota which participate in a variety of nervous systems.

Point 4: Line 48: I suggest to replace “microbiomes” with bacterial genera

Response 4:

Thank you for your comments, we have revised the word.

Line 50

are common bacterial genera microbiomes

Point 5: Line 91: I suggest to insert “with geographical location”

Response 5:

Thank you for your comments, we have inserted with geographical location.

Line 102

with geographical location location

Point 6: From line 183 to line 187: this sentence isn’t clear, please rewrite

Response 6:

I’m sorry for confusing you, the sentences were changed as follows:

Line 205

Fusobacterium, Propionibacterium, Haemophilus, and Streptococcus were the main bacteria in non-type 2 CRS patients. non-type 2 CRS, Fusobacterium, Propionibacterium, Hae-mophilus, and Streptococcus.

Point 7: Figure 1: Propionibacterium acnes and Haemophilus haven’t an artwork, why? If it is possible add a picture or similar

Response 7:

I’m sorry but I think you did not notice the Propionibacterium acnes and Haemophilus in the figure 1. The artwork of Propionibacterium acnes is in the upper box next to the S. aureus. And the artwork of Haemophilus is in the right box next to the Veillonella.

Point 8: Lines 84, 85, 87, 88, 89, 91, 94, 96, 115, 116, 137, 138, 139, 141, 142, 143, 180, 181, 186, 187, 188, 191, 193, 227, 238, 239, 240, 241, 243, 252, 267, 269, 270: pay attention to put all the bacteria in italic

Response 8:

Thanks for your suggestion and we have corrected all of them to italic.

Sincerely,

Tae Hoon Kim, M.D., PhD.(corresponding author)

Professor of Otorhinolaryngology-Head and Neck Surgery

Korea University College of Medicine

Director, External Communication/Cooperation Team, Anam Hospital

Director, International Medical Device Clinical Trial Support Center,

Korea University Medicine 73, Goryedae-ro, Seongbuk-gu, Seoul 02841, Korea

73, Goryeodae-ro, Seongbuk-gu, Seoul 02841, Korea

TEL: (82)-2-920-6405, (82)-10-9491-9886
FAX: (82)-2-925-5233
E-mail: [email protected]

Reviewer 2 Report

The authors summarized the previous reports in a concise manner. I feel it would be helpful for potential readers to understand the role of microbiome in nasal mucosal diseases. However, there are several points that could improve the manuscript as followed.

  1. The authors mentioned the general information or gut microbiota in the introduction section. Intro may focus on the reasons why the association between microbiota and nasal or respiratory mucosal diseases needs to be considered.
  2. In line 161-162, the authors decribed that non type 2 CRS showed a lack of nasal polyps. However non-eosinophilic CRS has also been accompanied by nasal polyps frequently especiall in Asian countries. It needs to be revised.
  3. Some recent pubications related to this topic are omitted. For example, 

     Allergy Asthma Immunol Res. 2021;13:e46.

     https://doi.org/10.1007/s00405-021-06747-z

     https://doi.org/10.1371/journal.pone.0249688

Author Response

Point 1: The authors mentioned the general information or gut microbiota in the introduction section. Intro may focus on the reasons why the association between microbiota and nasal or respiratory mucosal diseases needs to be considered.

Response 1:

Thank you for your valuable comments, We changed the paragraph as follows.

Line 55-63

Human nasal mucosa is the first contact point of inhaled environmental insults. Just as gut microbiota can protect the intestinal mucosa through immune regulation, microbi-ota in nasal mucosa are likely to play an important role in its mucosal immunity. Alt-hough much research has been conducted concerning the role of microbiota in lower res-piratory tract disorders such as asthma, the role of microbiota in the upper respiratory tract including human nasal mucosa has not been studied in detail especially in respect to immunity. Therefore, this review aims to clarify the relationships between different types of nasal mucosal diseases and microbiota in immunity, and introduces new technologies and methods to study microbiota.

Point 2: In line 161-162, the authors described that non-type 2 CRS showed a lack of nasal polyps. However non-eosinophilic CRS has also been accompanied by nasal polyps frequently especially in Asian countries. It needs to be revised.

Response 2:

Thank you for your comments, we have revised the sentence as follows

Line 180

Non-type 2 CRS is a heterogeneous disease characterized by a lack of nasal polyps (NPs)

Point 3: Some recent publications related to this topic are omitted.

Response 3:

Thank you very much for your advice. We have quoted these recent articles in this manuscript

Line 90

One study concluded that tissue samples were more suitable for assessing microbiological groups in CRS patients than nasal swabs, because they observed significant differences in the microbiota groups in the nasal swabs, while the differences observed in the tissue samples were smaller [30].

Line 94

But there are some discrepancies about the usefulness of two methods. [31]. Another data showed that tissue biopsy can’t provide additional information compared with multiple swab tests. In more than 90% of their cases, swabs from multiple sites provide compre-hensive information about patients' culturable pathogens.

Line 130

A most recent study in South Korea has obtained an interesting data [46]. They found that the use of antibiotics can cause differences in secretory proteome according to the condition of the disease. Their data suggest that the use of antibiotics should be considered as a con-founding factor in proteomics research.

Sincerely,

Tae Hoon Kim, M.D., PhD.(corresponding author)

Professor of Otorhinolaryngology-Head and Neck Surgery

Korea University College of Medicine

Director, External Communication/Cooperation Team, Anam Hospital

Director, International Medical Device Clinical Trial Support Center,

Korea University Medicine 73, Goryedae-ro, Seongbuk-gu, Seoul 02841, Korea

73, Goryeodae-ro, Seongbuk-gu, Seoul 02841, Korea

TEL: (82)-2-920-6405, (82)-10-9491-9886
FAX: (82)-2-925-5233
E-mail: [email protected]

Round 2

Reviewer 1 Report

None

Reviewer 2 Report

The authors have addressed my concerns successfully.